# Unsupervised Modality Adaptation in Human Action Recognition Via Cross-modal Representation Learning

**Abhi Kamboj**
Electrical and Computer Engineering
University of Illinois
akamboj2@illinois.edu

**Anh Duy Nguyen**
Computer Science
University of Illinois
duyan2@illinois.edu

**Minh Do**
Electrical and Computer Engineering
University of Illinois
minhdo@illinois.edu

## Abstract

Despite living in a multi-sensory world, most AI models are limited to textual and visual interpretations of human motion and behavior. In order to unlock the potential of diverse sensors, we investigate a method to transfer knowledge between modalities using the structure of a unified multimodal representation space for human action recognition (HAR). We introduce an understudied cross-modal transfer setting termed Unsupervised Modality Adaptation (UMA), where the modality used in testing is not used in supervised training. We develop three methods to perform UMA: Student-Teacher (ST), Contrastive Alignment (CA), and Cross-modal Transfer Through Time (C3T). Extensive experiments on various camera+IMU datasets demonstrate ST is effective on simple tasks, CA is the most modular and balanced method and C3T is the most robust through temporal noise. In particular, our C3T method introduces novel mechanics of aligning a signal across time-varying latent vectors, and we show that it demonstrates unique robustness to time-related noise, suggesting its potential for developing generalizable models for time-series sensor data.

## 1  Introduction

**Motivation:** Humans can naturally actuate a motion they have only seen before; however, transferring motion knowledge across sensors for machine learning models is nontrivial. Our interaction with computing has historically been centered around visual and textual modalities, which has provided these models an abundance of data. Thus, deep learning based human action recognition (HAR) systems often collapse 3D motion into related but imprecise modalities such as visual data [19, 35, 23, 40] or language models [32, 41, 37, 30, 11]. Sensing modalities in wearables (e.g. IMU, ECG, PPG, etc.) provide a salient signal to perform HAR, however, the data is less abundant and difficult to label.This raises the critical question of *how to integrate new sensors with existing ones in the absence of labeled data.* One promising solution is to leverage a well-documented modality to transfer knowledge to another modality, a process known as cross-modal transfer [28], ideally without additional human annotation effort. Existing cross-modal learning techniques assume semi-supervised or fully supervised settings. Cross-modal learning has not thoroughly been investigated in a setting where one modality is completely unlabeled during training. We refer to this as Unsupervised Modality Adaptation (UMA), similar to the widely used setting of Unsupervised Domain Adaptation [24] where the domain shift is a new modality.

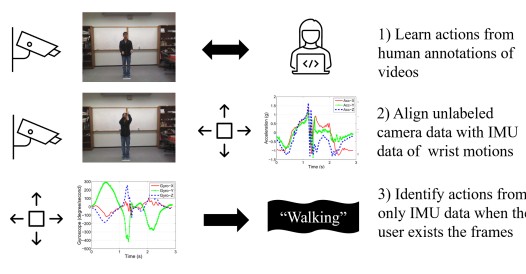

1) Learn actions from human annotations of videos

2) Align unlabeled camera data with IMU data of wrist motions

3) Identify actions from only IMU data when the user exists the frames

"Walking"

Figure 1: Motivation for Unsupervised Modality Adaptation (UMA)

Table 1: Data Splits for UMA

| Split | $X^{RGB}$ | $X^{IMU}$ | $Y$ | Size |
|---|---|---|---|---|
| Train a) $\mathcal{D}_{HAR}$ | ✓ | | ✓ | 40% |
| Train b) $\mathcal{D}_{Align}$ | ✓ | ✓ | | 40% |
| Val $\mathcal{D}_{Val}$ | | ✓ | ✓ | 10% |
| Test $\mathcal{D}_{Test}$ | | ✓ | ✓ | 10% |

**Contributions:** In order to perform UMA, we use the intuition that there exists some joint multi-modal representation space for HAR, that can be leveraged to infer the same action across different modalities. We propose 3 methods for this to extract and leverage this latent space. The first is a student teacher (ST) method akin to existing knowledge distillation methods for other domain adaptation or semi-supervised settings (Figure 2a). The second method performs contrastive alignment (CA) on latent representations of multimodal unlabeled data samples and uses a shared task head to perform transfer (Figure 2b). The third method extracts a time-varying latent dimension, i.e. a set of $t_{rec}$ latent vectors, and performs Cross-modal Transfer Through Time (C3T) (Figure 2c).

We test these methods on Inertial Measurement Units (IMUs) and RGB video data on four datasets. Although the ST method works best on simple datasets, CA performs better in more difficult visual tasks. This indicates latent space alignment captures hidden correlations allowing the model to leverage one modality to infer a structure in the other. Furthermore, C3T consistently performs the best and is the most robust to time-shift, misalignment, and time-dialation noise as it accounts for the temporal information within each data sample. This investigation of UMA cross-modal transfer lies at the intersection of transfer learning, multimodal representation learning and holds significant implications for the applicability of machine learning in more diverse, underexplored, modalities.

## 2   Methods

We construct the Unsupervised Modality Adaptation (UMA) setting with RGB videos as the source of labeled data and IMU data as the target unlabeled modality. As shown in Figure 1, a system can easily record synchronous data between these modalities, and then leverage an RGB model to perform HAR with only the IMU data. We mimic this setting by dividing 4 existing multimodal datasets into 4 splits as shown in Table 1. Training for each method occurs in two phases: *a) Supervised Learning* with RGB data on $\mathcal{D}_{HAR}$ and *b) Unsupervised Alignment* across both modalities on $\mathcal{D}_{Align}$. Inference can occur, with any combination of the input modalities (Table 2), however, we focus testing on IMU data (Table 3). $\mathcal{D}_{Val}$ was used for hyperparameter search during training, and all tables report the average accuracy over three trials on $\mathcal{D}_{Test}$. We propose three methods for transferring knowledge to a new sensing modality without exposure to labels in that modality, as depicted in Figure 2:

**(1) Student-Teacher (ST) Figure 2a:** The ST method leverages an RGB video model trained in phase a) to produce psuedo-labels to train the IMU model in phase b). In this case, the latent space is the output logit space aligned using the cross-entropy loss, Equation (1). Various student-teacher models have been proposed [21, 39, 3, 36]; however, these models often assume the availability of student-teacher-labeled modality pairs during training to distill knowledge from the teacher to the student. Furthermore, we use only one student and teacher module distinguishing our method from Thoker and Gall [36] who require an ensemble to strengthen the model in a similar setting.

**(2) Contrastive Alignment (CA) Figure 2b:** The CA method performs phase a) Supervised RGB training in the same fashion as the student teacher, however, it uses a model with 2 parts: An encoder $f^{(1)}$ to extract the latent variable $z$, and a task specific MLP head $h$. The extracted latent space $\mathcal{Z}$ allows for scalability and interoperability of adding different sensing modalities, types of encoders, and output task heads. Phase b) performs unsupervised contrastive alignment with the outputs of the the RGB encoder $f^{(1)}$ and the IMU encoder $f^{(2)}$ on unlabeled data using the symmetric contrastive loss formulation from [32, 27, 13] given by Equation (2), further detailed in appendix. The symmetric contrastive loss will cluster representations in $\mathcal{Z}$ by cosine similarity, which brings about the desired property of the latent space that vectors of the same class are near each other. A unified representation

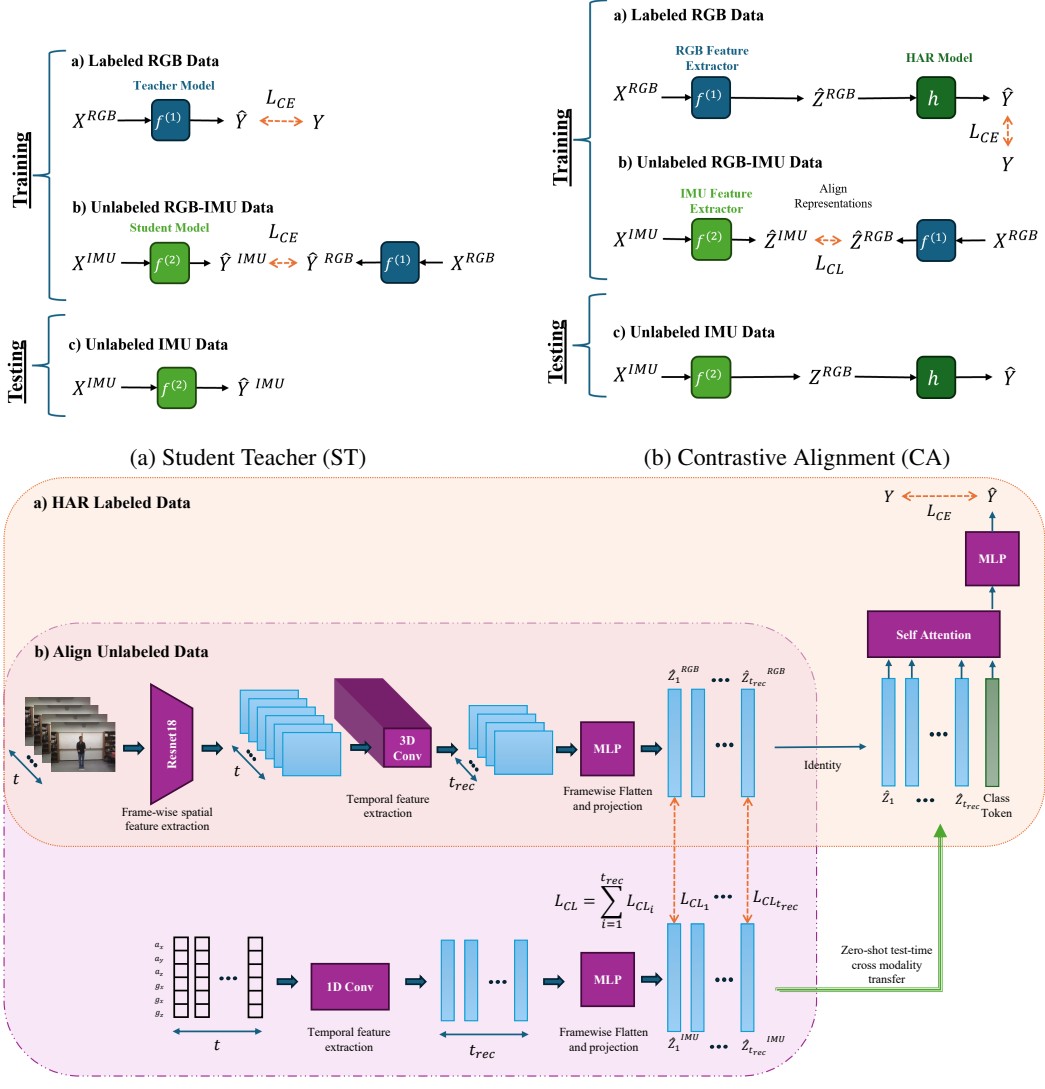

(a) Student Teacher (ST)

(b) Contrastive Alignment (CA)

(c) Cross-modal Transfer Through Time (C3T)

Figure 2: Training and testing for three methods leveraging a unified latent space for UMA.

space for separate modalities allows the decision boundary trained on RGB representations, $h$, to be used to recognize actions on IMU representations. This latent space is visualized in Figure 3.

**(3) Cross-modal Transfer Through Time (C3T) Figure 2c:** ST and CA do not leverage the temporal information of the data, making them difficult to use in real-world settings. C3T removes the final linear layer from the feature encoders of CA and uses the output of the temporal convolutions directly. This temporal receptive field would have extracted the salient features of neighboring time steps of the data. Then during the alignment phase, each of these time vectors is aligned with the same time vector from the other modality, using the same contrastive loss CA uses (Equation (2)). When training the HAR model, we use self attention with a learned class token to predict the action, similar to the ViT architecture [9], but instead of inputs being image chunks, they are temporal feature chunks. The intuition is that the encoder will learn which tokens over time are the most informative for the action class and predict accordingly.

## 3 Experiments

***How do we train the CA and C3T Architectures?*** We experimented with four ways to perform the two phases of training, as shown in Table 2. *1) Align First:* First aligns the representations

Table 2: **Additional Experiments**: Performance of ST, CA, and C3T across various training methods, modalities, and noise. All results report UMA accuracy on IMU data, except modality test 2. and 3.

| Model | Training | | | | Modality Testing | | | Noise Experiments | | | | |
|---|---|---|---|---|---|---|---|---|---|---|---|---|
| | 1. Align | 2. HAR | 3. Inter | 4. Comb | 1. IMU | 2. RGB | 3. Both | 1. Crop | 2. Misalign | 3. Dilate | 4.All | None |
| ST | - | - | - | - | 12.9 | 53.8 | 17.0 | 3.4 | 5.7 | 5.7 | 10.2 | 12.9 |
| CA | 38.6 | 43.2 | 27.3 | **42.6** | 42.6 | 56.8 | 60.2 | 10.2 | 2.3 | 21.6 | 18.2 | 42.6 |
| C3T | **62.5** | 35.2 | 51.1 | 27.9 | **62.5** | 78.4 | 79.5 | 52.3 | **46.6** | **56.8** | **58.0** | 62.5 |

Table 3: **UMA vs. Supervised Performance:** The modules $f^{(1)}$, $f^{(2)}$, and $h$ can operate in supervised or UMA (ST, CA, CT3) modes. Top-1 and Top-3 accuracies shown.

| | Model | UTD-MHAD | | CZU-MHAD | | MMACT | | MMEA-CL | |
|---|---|---|---|---|---|---|---|---|---|
| | | Top-1 | Top-3 | Top-1 | Top-3 | Top-1 | Top-3 | Top-1 | Top-3 |
| Supervised | IMU | **87.9** | **97.7** | **95.1** | 98.2 | 70.0 | 90.0 | 65.8 | 87.6 |
| | RGB | 53.8 | 73.1 | 94.0 | **99.7** | 42.1 | 61.6 | 54.2 | 77.1 |
| | Fusion | 62.5 | 82.2 | 95.0 | 98.5 | **76.7** | **92.0** | **80.1** | **92.7** |
| UMA | Random | 3.7 | 11.1 | 4.6 | 16.6 | 2.9 | 8.6 | 3.1 | 9.4 |
| | ST | 12.9 | 24.6 | 41.1 | 61.9 | 17.6 | 34.7 | 9.9 | 22.7 |
| | CA | 42.6 | 67.4 | 70.0 | 92.7 | 24.5 | 47.6 | 29.3 | 51.7 |
| | C3T | **62.5** | **86.4** | **84.2** | **96.7** | **32.4** | **57.9** | **51.2** | **78.8** |

generated by the RGB and IMU encoders on $\mathcal{D}_{Align}$, then freezes the RGB encoder and trains the HAR module on $\mathcal{D}_{HAR}$. 2) HAR First: First trains the HAR module on $\mathcal{D}_{HAR}$, then freezes the RGB encoder and performs cross-modal alignment on $\mathcal{D}_{Align}$. 3) Interspersed Training: Intermittently learns one epoch from $\mathcal{D}_{Align}$ then one epoch from $\mathcal{D}_{HAR}$. 4) Combined Loss: Zips the $\mathcal{D}_{HAR}$ and $\mathcal{D}_{Align}$ dataloaders, computes gradients of the model for each batch, and updates the model with the combined loss $L_{Total} = L_{CE} + L_{CL}$. Training method 1) performed the best for C3T and 4) for CA.

We hypothesize that in method 2) training the HAR model first yields a latent space to capture the best HAR features for RGB data, which is not directly applicable to IMU data. Method 3), faced instability in training and was unable to converge. Method 4), performed better than 1) for CA potentially since one loss acted as a regularizer for the other pushing the latent space $\mathcal{Z}$ to the ideal balance for cross-modal transfer in HAR. The other experiments in this work use training method 1).

***Can UMA methods retain performance on the original modality they were trained on? Can they leverage both modalities?*** Table 2 shows the result of training in the UMA setting, but testing with all combinations of the modalites: *1. RGB* uses the RGB encoder and HAR module, *2. IMU (Zero-Shot Transfer)* uses the IMU encoder and the HAR module, and *3. Fusion* performs feature-level fusion (or late fusion for the ST method), by averaging the outputs of each of the encoders. Surprisingly for C3T, fusing (highlighted in red) performs better than RGB, indicating that unlabeled IMU data may add structure to the shared latent space to boost performance.

***Can our methods perform well on diverse RGB-IMU datasets?*** We test on a small yet structured dataset (UTD-MHAD [8]), a slightly larger dataset captured in a controlled environment with dense inertial data, i.e. 10 on-body sensors (CZU-MHAD [7]), one very large dataset with viewpoint and occlusion challenges (MMACT [21]), and one large egocentric camera dataset (MMEA-CL [43]). More details on the baselines and datasets are given in Appendix D.2 and Appendix D.1 respectively. The results (Table 3) show large performance gaps indicating C3T is far superior to the other methods. Furthermore, we compare the UMA performance against the Supervised methods and show C3T even outperforms supervised RGB on some datasets where the IMU data is highly informative.

***How robust is our method to time shifts and noise?*** Table 2 demonstrates C3T's superiority in the presence of temporal noise. We test on three real life noisy scenarios: 1) Crop: Accounts for the continuous nature of real-time action recognition (i.e. there is no defined start and stop time) by randomly shifting and cropping the time sequence of both modalities by 60% 2) Misalign: Imitates slight hardware asynchrony or different device framerates of the multimodal system by performing crop but on only one of the modalities. 3) Dilation: Mimics an action being performed slower, by randomly cropping both modalities to 50% of their original size an then upsampling.

**Qualitative Results:** We visualized the latent space outputs of the CA model using TSNE plots (Figure 3). These plots show training when the alignment phase (phase b in Figure 2) is performed first, and then labeled-RGB training (phase a) is performed. The model quickly segments classes

during the align phase, even without labels, suggesting that the data's natural structure facilitates class distinction across different modalities. This implies that our methods could potentially adapt to new class labels during testing with just a few samples, as the latent structure would have already grouped similar classes. Furthermore, after alignment and HAR training, we notice how the model tends to misclassify points that are near the boundary between clusters. These visualizations support our initial hypothesis (Figure 1) on how a joint latent space could be leveraged to effectively perform UMA, by using a classification head trained only on RGB data.

Interestingly, we observed that IMU data points consistently cluster towards the center of the plot, with RGB points surrounding them. This pattern persists even in early alignment stages, suggesting it's not solely due to labeled RGB HAR training. While this might indicate that RGB data is more informative, it contradicts our quantitative findings where supervised IMU models outperform RGB models for our given datasets. This phenomenon warrants further investigation as it may have implications for continual learning, test-time adaptation, or domain adaptation, where different modalities should be leveraged differently depending on their placement in the shared latent space.

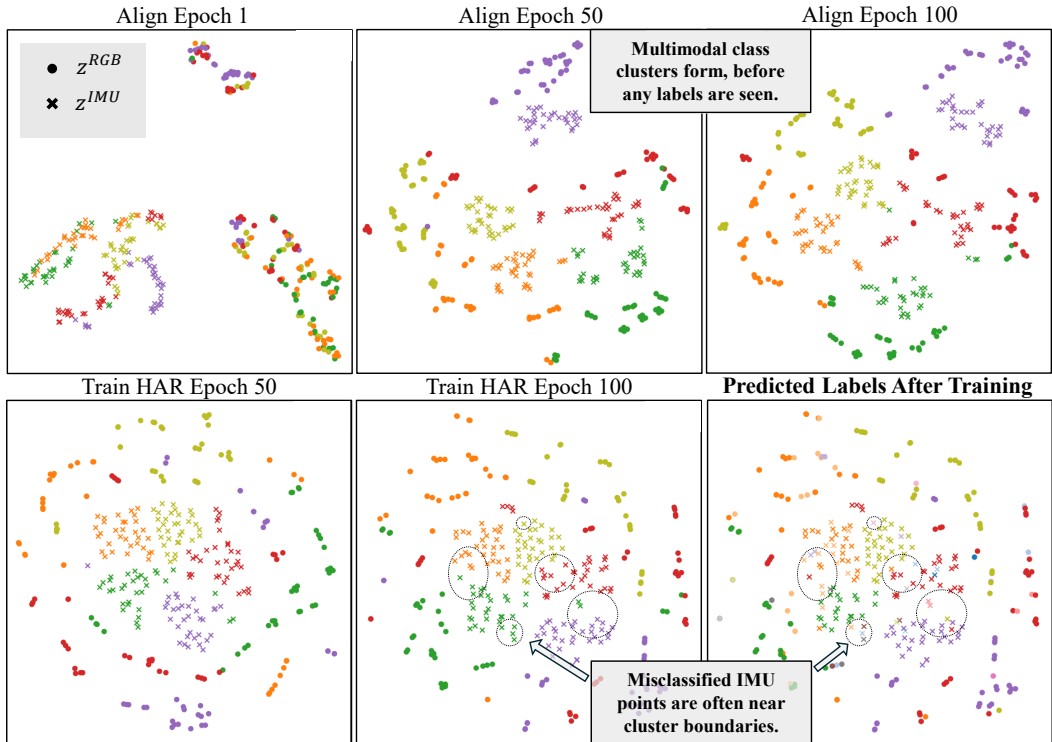

Figure 3: **CA TSNE Plots in UMA Training Method 1:** The following shows the progression of the latent representations of datapoints for 5 classes (Bowling, Clap, Draw circle (clockwise), Jog, Basketball shoot) during training CA on the UTD-MHAD dataset. At the end we plot the predicted labels and circle areas of confusion, which seems to often occur at the boundaries between clusters.

## 4  Conclusion:

In this paper, we motivated and explored the Unsupervised Modality Adaptation (UMA) framework for human action recognition which challenges models to perform inference with a modality that is unlabeled during training. We conduct experiments to determine how to construct a unified latent space between modalities, outline three methods to perform UMA with their constructed latent space and compare their strengths in various settings. We hope our results inspire others to exploit cross-modal latent spaces to integrate continuous time sensor signals into AI models for more robust human motion understanding.

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

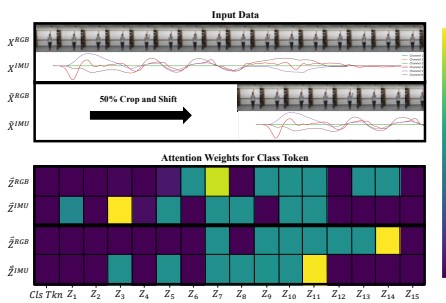

Figure 4: **Attention heatmap for C3T's HAR module:** Input shifts shift the attention weights of the temporal latent vectors.

Table 4: **C3T HAR Module Ablations:** Comparison of 2 Attention methods and two MLP methods.

| Input | Attention | | MLP | |
|---|---|---|---|---|
| | Class | Concat | Add | Concat |
| Clean | 62.5 | 44.32 | 56.82 | **70.45** |
| Noisy | **52.27** | 43.18 | 50.00 | 43.18 |

Table 5: **Architecture Ablation:** Comparison of different architectures for RGB and IMU encoders across methods. Encoder types are reported as (RGB-Spatial / RGB-Temporal / IMU-Temporal), where C = Convolutional and A = Attention.

| Method | Encoders | Params (M) | Acc. (%) |
|---|---|---|---|
| ST | C/C/C | 129.2 | **12.9** |
| | C/C/A | 97.8 | 10.2 |
| | C/A/C | 871.2 | 11.4 |
| | A/C/C | 291.5 | 5.7 |
| CA | C/C/C | 163.8 | **38.6** |
| | C/C/A | 132.3 | 19.3 |
| | C/A/C | 905.7 | 34.1 |
| | A/C/C | 326.0 | 26.1 |
| C3T | C/C/C | 137.7 | **62.5** |
| | C/C/A | 106.3 | 15.9 |
| | C/A/C | 879.6 | 53.4 |
| | A/C/C | 300.0 | 33.0 |

## A    Appendix / supplemental material

### A.1    Additional Experiments

**Ablations:**

We conducted comprehensive ablation experiments on our model architecture (Table 5), comparing convolutional and attention modules for RGB (spatial and temporal) and IMU (temporal) encoders. Results generally favored convolutional architectures across various methods in our UMA setting. Notably, C3T's superior performance cannot be attributed solely to its attention head leveraging temporal information, as ST or CA models with temporal attention did not perform comparably well. Instead, C3T's effectiveness stems from its unique method of alignment in the temporal space.

Further ablation on C3T head architectures (Table 4) compared the class token-based self-attention head with alternatives: concatenating output attention tokens and projecting, adding latent vectors $Z_1 \ldots Z_{t_{\text{rec}}}$ and passing through an MLP, and concatenating vectors and using an MLP. The latter two methods do not use attention. While concatenating latent vectors and using an MLP performed best on clean data, the class token attention mechanism offered superior robustness to noise. The attention visualization in Figure 4 corroborates these findings, showing the class token approach's resilience to shift noise. In addition, we notice all C3T variants outperformed CA and ST in UMA performance (Table 3) on the UTD-MHAD dataset, emphasizing C3T's strength in temporal alignment, regardless of the classification head.

## B    Background

We investigate the creation of a robust multi-modal latent space for human action recognition, denoted as $\mathcal{Z}$. We assume that there exists a learnable projection $f^{(k)}$ from every modality $k \in \{1 \ldots M\}$ to this latent space $\mathcal{Z}$, i.e. $f^{(k)} : \mathcal{X}^{(k)} \to \mathcal{Z}$, and there exists a learnable mapping $h$ from the latent space $\mathcal{Z}$ to the space of human actions $\mathcal{Y}$, i.e. $h : \mathcal{Z} \to \mathcal{Y}$.

The critical intuition that drives our method is that for a point $z_i \in \mathcal{Z}$, any $z_j$ "near" $z_i$ is likely to map to the same class as $z_i$, thus we can leverage the structure of $\mathcal{Z}$ to classify neighboring vectors regardless of which modality they are generated from, using the same decision boundary determined by $h$. In our experiments, we quantify closeness in terms of cosine similarity (and also perform some ablations with L2 distance metric).

For simplicity, we experiment with 2 modalities $M = 2$ and assume $n$ data points are split into 4 disjoint index sets $I_1 \cup I_2 \cup I_3 \cup I_4 \in \{1 \ldots n\}$. Under our cross-modal transfer setting, during

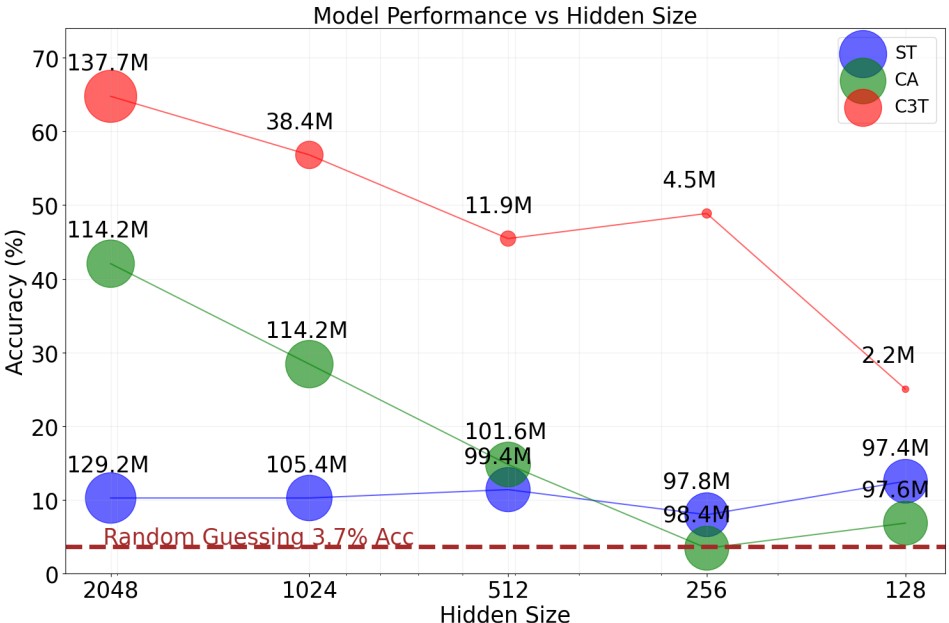

Figure 5: **Performance vs Model Size:** Bubbles show accuracy for each latent space hidden size. Bubble size indicates model parameters (millions).

training the model has access to 2 of these datasets. One contains labeled data for one modality $\mathcal{D}_{HAR} = \{(\mathbf{x}_i^1, \mathbf{y}_i)\}_{i=1}^{I_1}$ and the other contains pairs of data between the modalities but these points are unlabeled: $\mathcal{D}_{Align} = \{(\mathbf{x}_i^1, \mathbf{x}_i^2)\}_{i=1}^{I_2}$. This is analogous to having an existing sensor with labeled data, and introducing a new sensor in which data can be synchronously collected, but there is no additional annotation effort. Now, if the sensor with labeled data is not present, can the model still perform inference with the unlabeled modality? The third and fourth sets are used for validation and testing and contain only labeled data from the second modality, i.e. $\mathcal{D}_{Val} = \{(\mathbf{x}_i^2, \mathbf{y}_i)\}_{i=1}^{I_3}$ and $\mathcal{D}_{Test} = \{(\mathbf{x}_i^2, \mathbf{y}_i)\}_{i=1}^{I_4}$.

## C  Methods

We propose three methods for transferring knowledge to a new sensing modality without exposure to labels in that modality: (1) a student-teacher approach, (2) a contrastive alignment technique, and (3) a contrastive alignment through time method. Given our motivation of human action recognition, we experiment with RGB videos as the source of labeled data $x^{(1)} = x^{(RGB)}$ and IMU data as the target data $x^{(2)} = x^{(IMU)}$.

Training for each method occurs in two phases: a) Supervised Learning with RGB data and b) Unsupervised alignment across both modalities. Inference can occur, with any combination of the input modalities or both, as shown in Table 2, however, we mainly focus on the results of the IMU test, shown extensively in Table 6. All tables report the Top-1 accuracy on the $\mathcal{D}_{Test}$ for each method is given by: accuracy $= \frac{1}{M} \sum_{i=1}^M \mathbb{1}_{\hat{\mathbf{y}}_i = \mathbf{y}_i}$ To ensure the robustness and reliability of our empirical results, we conducted each experiment three times using different random seeds. The reported accuracies are averaged across these trials, providing a more rigorous and representative assessment of the model's performance.

**Model Architecture:**

Each of the methods below uses a combination of the following three neural network modules: *(i)Video Feature Encoder* $f^{(1)} : \mathcal{X}^{(1)} \to \mathcal{Z}$: This module applies a pretrained Resnet18 to every frame a video and then performs a single 3D convolution and a simple 2-layer feed forward network (FFN) with ReLU activations. (ii) *IMU Feature Encoder* $f^{(2)} : \mathcal{X}^{(2)} \to \mathcal{Z}$: This module consists of a 1D CNN followed by a FFN. (iii) *HAR Task Decoder* $h : \mathcal{Z} \to \mathcal{Y}$: This is a simple FFN.

**Implementation Details:** $\mathcal{D}_{Val}$ was used to perform a minor hyperparameter search on the UTD-MHAD [8] dataset and testing was only done once on the best chosen model. The methods the performed best with a learning rate of $1.5e-3$ and a latent representation dimension of $1024$. All experiments were ran on either a single 10GB NVIDIA GeForce 3080 or a single 16GB NVIDIA Quadro RTX 5000, and the exact length of the experiments varied per baseline and dataset. The model was trained in Pytorch using an Adam optimizer, the learning rate was empirically determined and the loss functions defined above.

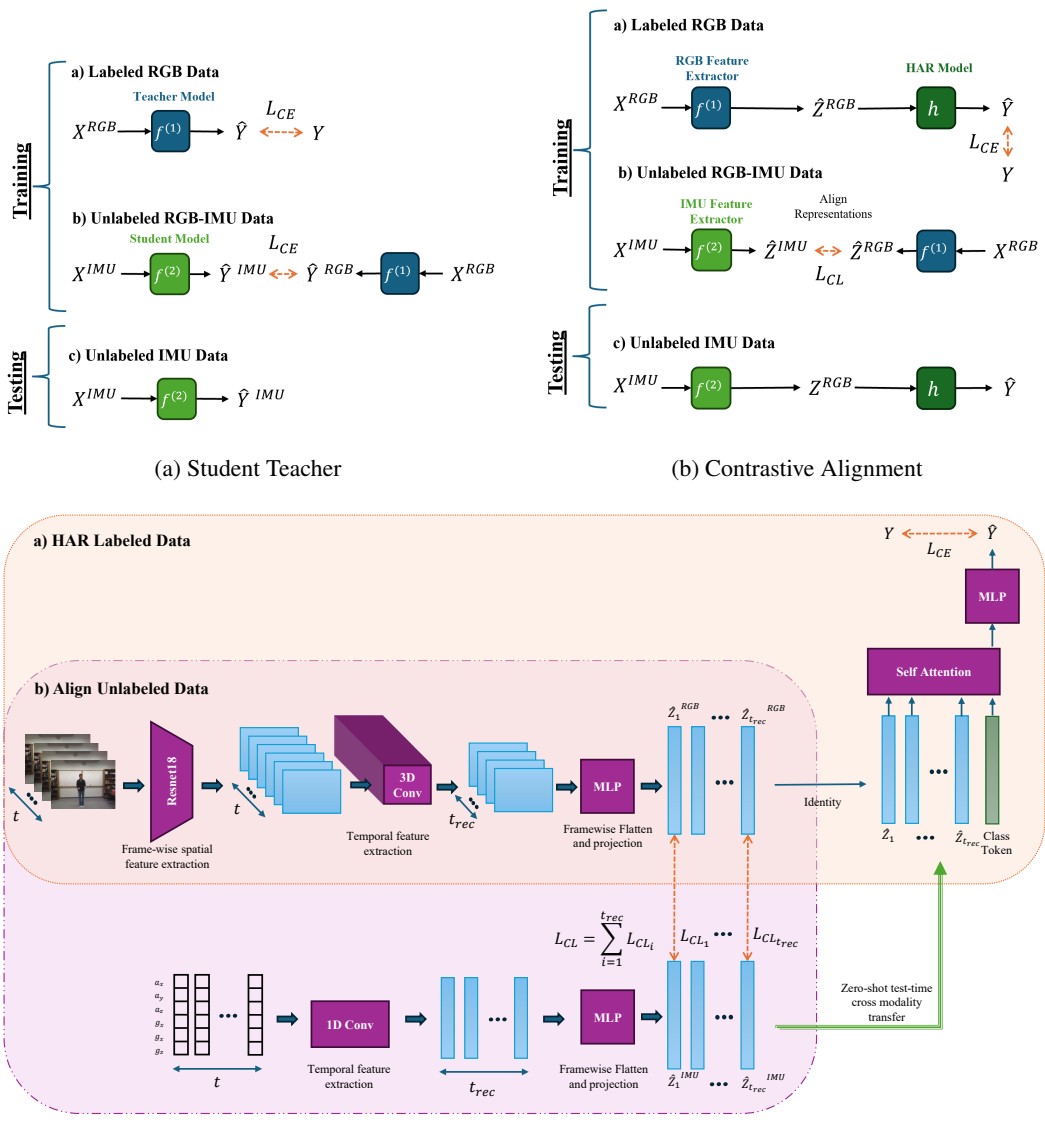

(a) Student Teacher

(b) Contrastive Alignment

(c) Cross-modal Transfer Through Time

Figure 6: Training happens in two phases: a) trains the HAR model on labeled RGB inputs and b) aligns unlabeled IMU and RGB modalities. The student-teacher method (a) trains a model with the teacher modality in a) and uses that model to psuedolabel and train the student model in (b). The Contrastive Alignemnt (CA) method trains a feature extractor and task module in a) aligns each modality's feature extractor in b). (c) Cross-modal transfer through time method is similar to CA but aligns representations across time in the latent space and uses self-attention across the time sequence to perform HAR inference. All models are tested for zero-shot cross-modal transfer to the IMU modality.

## C.1 Student Teacher:

The student teacher method leverages a trained video model to produce psuedo-labels to train the IMU model. In this case there is no intermediate latent space and the model maps each modality's encoder directly to the output task. This essentially uses cross-entropy to align the models directly in the label space: $\mathcal{Z} = \mathcal{Y}$.

Various student teacher models have been proposed [21, 39, 3, 36]. However, these models often assume the availability of student-teacher-labeled modality pairs during training to distill knowledge from the teacher to the student when updating the corresponding losses. Furthermore, we use only one student and teacher module distinguishing our method from Thoker and Gall [36] who require an ensemble of to strengthen the model in a similar setting.

We denote the teacher network as $f^{(1)} : \mathcal{X}^{(1)} \to \mathcal{Y}$ and the student network as $f^{(2)} : \mathcal{X}^{(2)} \to \mathcal{Y}$. First, we train the teacher $f^{(2)}$ on $\mathcal{D}_{HAR}$ with labeled RGB data. Next, in order to train $f^{(1)}$ on $\mathcal{D}_{Align}$, we first use $f^{(2)}(x_i^{(1)}) = \hat{y}_1$ to generate psuedo-labels for every datapoint $i \in I_2$. Then we use the augmented dataset $\hat{\mathcal{D}}_{Align} = \{(\mathbf{x}_i^1, \mathbf{x}_i^2, \hat{\mathbf{y}}_i)\}_{i=1}^{I_2}$ to train $f^{(1)}$.

Both student and teacher networks were trained using cross-entropy loss:

$$L_{CE}(P_{\hat{y}}, P_y) = -\frac{1}{N} \sum_{i=1}^{N} \sum_{j=1}^{C} \mathbb{1}_{y_i=j} \log\left(\frac{\exp \hat{y}_{i,j}}{\sum_{i=1}^{M} \exp \hat{y}_{i,j}}\right) \tag{1}$$

where $\hat{y}_i = f(x_i)$ is the output of the $i$th sample in the batch of $N$ samples, $\hat{y}_{i,j}$ is the score for the $j$th class out of $C$ classes, and $P_y$ represents the probability distribution produced by a given model's output logits. The teacher network minimizes $L_{CE}(P_{f^2(x)}, P_y)$ student network minimizes $L_{CE}(P_{f^1(x)}, P_{f^2(x)})$. Since the student approximates the teacher and the teacher approximates the true distribution, this implies that the student can only be as good as the teacher at approximating the true distribution: $L_{CE}(P_{f^1(x)}, P_y) > L_{CE}(P_{f^2(x)}, P_y)$. Thus given noise in the teacher distribution the student also suffers in performance, as shown in our experiments section. Furthermore, given that the latent space is the label distribution, this method provides little flexibility in extending to various modalities and tasks.

## C.2 Contrastive Alignment:

The contrastive alignment method performs phase a) Supervised RGB training in the same fashion as the student teacher, however, it uses an a model with 2 parts: An encoder $f^{(1)}$ to extract the latent variable $z$, and a task specific MLP head $h$. The extracted latent space $\mathcal{Z}$ allows for scalability and interoperability of adding different sensing modalities, types of encoders, and output task heads.

Phase b) now performs unsupervised contrastive alignment with the outputs of the the RGB encoder $f^{(1)}$ and the IMU encoder $f^{(2)}$ on unlabeled data. To align different modalities in the feature space on $\mathcal{D}_{Align}$ we use a symmetric contrastive loss formulation $L_{CL}$ [32, 27, 13] with temperature parameter $\tau$:

$$L_{CL} = -\frac{1}{N} \sum_{i=1}^{N} \log \frac{\exp(\langle \hat{z}_i^{(1)}, \hat{z}_i^{(2)} \rangle / \tau)}{\sum_{i=1}^{N} \exp(\langle \hat{z}_i^{(1)}, \hat{z}_i^{(2)} \rangle / \tau)}, \text{where } \hat{z}_i^{(k)} = \frac{f^{(k)}(x_i^{(k)})}{||f^{(k)}(x_i^{(k)})||}, k \in \{1, 2\} \tag{2}$$

The symmetric contrastive loss will cluster representations in $\mathcal{Z}$ by cosine similarity, which brings about the desired property of the latent space that vectors of the same class are 'near' each other. This method clusters similar representations in the latent space despite being from separate modaliteis, and thus the decision boundary trained on RGB representations, $h$ can be used to recognize actions on IMU represntations.

## C.3 Contrastive Alignment Through Time

The previous two discussed methods do not leverage the temporal information in time continuos data signals, which make them difficult to use in real-world settings, where there may be temporal noise (shift/misalignment) and the actions may occur over longer or shorter periods of time. We thus propose a Cross-modal Transfer Through Time (C3T) model that leverages the temporal information of sensing modalities when aligning and fusing their representations. C3T remove's the FFN from the feature encoders and use the output of the temporal convolutions directly. This temporal receptive field would have extracted the salient features of neighboring time steps of the data. We use each of these time steps as the a $z_t$ latent vector. Then during the alignment phase, we align each of these time vectors with the same time from the other modality. When training the HAR model,

Table 6: UMA from the RGB to IMU sensor modalities. No existing methods are easily adapted to this task, and the performance of all the proposed methods vary greatly.

| Model | UTD-MHAD | MMACT | MMEA- CL | CZU-MHAD |
|---|---|---|---|---|
| Sensor Fusion (2019) [42] | 5.2% | 3.2 % | 4.1 % | 4.5 % |
| HAMLET (2020) [17] | 4.6 % | 3.2 % | 4.1 % | 4.5 % |
| ImageBind (2023) [13] | 11.3 % | 3.34 % | 40.1 % | 4.54 % |
| Student Teacher (Ours) | 61.6% | 17.1 % | 9.13 % | **93.9%** |
| Contrastive Alignement (Ours) | **65.9%** | **33.7 %** | 42.7 % | 77.2 % |
| Contrastive Alignment Through Time (Ours) | 63.6% | 20.6 % | **47.5 %** | 80.3 % |

we use self attention with a learned class token to predict the action The intuition is that the encoder will learn which tokens over time are the most informative for the action class and predict accordingly. This is a common method to perform classification with transformers [9, 4, 26]. Our implemented HAR task head is most notably similar to the ViT architecture [9], but instead of inputs being image chunks, they are temporal feature chunks.

The updated modules are as follows:

**Video Feature Encoder** $f^{(1)} : \mathcal{X}^{(1)} \to \mathcal{Z}^{t_{rec}}$: This module applies a pretrained Resnet18 to every frame a video and then performs a single 3D convolution. The resulting output is a set $t_{rec}$ z: $\hat{\mathbf{Z}}^{(1)} = (\hat{z}_1^{(1)} \ldots \hat{z}_{t_{rec}}^{(1)})$.

**IMU Feature Encoder** $f^{(2)} : \mathcal{X}^{(2)} \to \mathcal{Z}^{t_{rec}}$: This is a 1D CNN that decreases the time dimension to $t_{rec}$, resulting in an output of $\hat{\mathbf{Z}}^{(2)} = (\hat{z}_1^{(2)} \ldots \hat{z}_{t_{rec}}^{(2)})$.

**HAR Task Decoder** $h : \mathcal{Z}^{t_{rec}} \to \mathcal{Y}$: This module is like a transformer encoder that uses self-attention on an input sequence of length $t_{rec}$ vectors appended with a learned class token. The output class token of the self attention layer is then passed through a FFN and outputs a single action label $y_i$.

# D   Results

## D.1   Datasets

We present results on small yet structure dataset (UTD-MHAD), one larger dataset captured in a controlled environment (CZU-MHAD), one very large dataset with various challenges (MMACT), and one egocentric camera dataset (MMEA-CL). For each of these datasets we create an approximately 40-40-10-10 percent datasplit for the $\mathcal{D}_{Align}$, $\mathcal{D}_{HAR}$, $\mathcal{D}_{Val}$, and $\mathcal{D}_{Test}$ splits respectively as shown in Table 7.

**UTD-MHAD** Most of the development and experiments were performed on the UTD-Multi-modal Human Action Dataset (UTD-MHAD) [8]. This dataset consists of roughly 861 sequences of RGB, skeletal, depth and an inertial sensor, with 27 different labeled action classes performed by 8 subjects 4 times. The inertial sensor provided 3-axis acceleration and 3-axis gyroscopic information, and all 6 channels were used for in our model as the IMU input. Given our motivation, we only use the video and inertial data; however, CA can easily be extended to multiple modalities.

**CZU-MHAD** The Changzhhou MHAD [7] dataset provides about 1,170 sequences and includes depth information from a Kinect camera synchronized with 10 IMU sensors, each 6 channels, in a very controlled setting with a user directly facing the camera. We concatenate the IMU data to provide a 60-channel input as the IMU modality and use depth as the input modality. Given the controlled environment and dense IMU streams, the models performed the best on this dataset.

**MMACT** The MMAct dataset [21] is a large scale dataset containing about 1,900 sequences of 35 action classes from 40 subjects on 7 modalities. This data is challenging because it provides data from 5 different scenes, including sitting a desk, or performing an action that is partially occluded by an object. Furthermore, the data was collected with the user facing random angles at random times. The dataset contains 4 different cameras at 4 corners of the room, and it measures acceleration on the user's watch and acceleration, gyroscope and orientation data from a user's phone in their pocket. We only use the cross-view camera 1 data, and again we concatenate the 4 3-axis inertial sensors into one 12 channel IMU modality.

**MMEA-CL** The multimodal egocentric activity recognition dataset for continual learning (MMEA-CL) is a recent dataset motivated by learning strong visual-IMU based representations that can be used for continual learning. It provides about 6,4000 samples of synchronized first-person video clips and 6-channel accelerometer and gyroscope data from a wrist worn IMU. The dataset's labels features realisitc daily actions in the wild, as opposed to recorded sequences in a lab. Due to issues with the data and technical constraints, we downsize the data proportionally and use about 1,000 samples. Nonethless, CA's superior performance shows how this method can generalize to a different camera view, and different types of activities.

## D.2 Baselines

Many works deal with robustness to missing sensor data during training or testing, however, few works deal with zero-labeled training data from one modality. As a result, constructing baselines was tricky and most methods had to be modified or adopted to fit our our approach.

### D.2.1 Student Teacher Baseline

Various student teacher models have been proposed [21, 39, 3]. However, these models often assume the availability of student-teacher labeled modality pairs during training to distill knowledge from the teacher to the student when updating the corresponding losses. Thus most of these architectures are not directly applicable. Nonetheless, we borrow the basic concept of a teacher modality distilling knowledge to the student modality through psuedo labels, most similar to [36].

We denote the teacher network as $g^{(t)} : \mathcal{X}^{(1)} \to \mathcal{Y}$ and the student network as $g^{(s)} : \mathcal{X}^{(2)} \to \mathcal{Y}$. Our designated student-teacher baseline uses $\mathcal{D}_{HAR}$ to train $g^{(t)}$ on the RGB data. Next, in order to train $g^{(s)}$ on $\mathcal{D}_{Align}$, we first use $g^{(t)}(x_1^{(1)}) = \hat{y}_1$ to generate psuedo-labels for every datapoint $i \in I_2$. Then we use the augmented dataset $\hat{\mathcal{D}}_{Align} = \{(\mathbf{x}_i^1, \mathbf{x}_i^2, \hat{\mathbf{y}}_i)\}_{i=1}^{I_2}$ to train $g^{(s)}$. We note that $g^{(t)}$ and $g^{(s)}$ have similar architectures to $h(f^{(1)}(\cdot))$ and $h(f^{(2)}(\cdot))$, respectively for a fair performance comparison.

This student teacher baseline presents a strong solution for our setup, and gives a competitive performance in Table 6. However, the only method it outperforms CA was was in the relatively easy CZU-MHAD dataset. One drawback, with this model is that it requires labeled data from both modalities to improve it's performance, whereas CA can use unlabeled data to learn some correlation between modalities in $\mathcal{Z}$. This representation space can be used for other purposes as well, such as dynamically adding modalities or task specific heads.

### D.2.2 Sensor Fusion Baselines

Many IMU-RGB based sensor fusion models have the ability to train on partially available or corrupted data and are robust to missing modalities during inference [18, 17]. No works have attempted the extreme case where one modality is completely unlabeled during training. Existing esnsor fusion methods can be adapted to our setup using a psuedo- labeling technique, similar to the student-teacher model above. The difference here is that the model learns a joint distribution between the two modalities so hopefully it may be able to learn some correlation between the models. Nonetheless, we show that these methods cannot generalize to the scenario where there is zero-labeled training data for one modality.

Let $g(\cdot, \cdot) : (\mathcal{X}^{(1)}, \mathcal{X}^{(2)}) \to \mathcal{Y}$. Our approach uses $\mathcal{D}_{HAR}$, to train by passing in zeros for one modality, e.g. we train $g(\cdot, \mathbf{0}) : \mathcal{X}^{(1)} \to \mathcal{Y}$. Then, with $\mathcal{D}_{Align}$ we use $g(\cdot, \mathbf{0})$ to generated psuedo-labels and then train $g(\mathbf{0}, \cdot, )$ with those labels.

We reproduced the conventional sensor fusion models (early, feature, and late) from [42] and indicate the performance of the top model on 6. We further reproduce a self-attention based sensor fusion appraoch (HAMLET [17]) and tested it on our setup. We selected these model due to their state-of-the-art performance on the UTD-MHAD dataset, making them ideal benchmarks for comparison with our model.

**More details on developing sensor fusion baselines:**

Given that there is no open source implementation for the HAMLET attention based sensor fusion method [17], we reproduce it from scratch. We follow a very similar architecture; however, extract spatio-temporal results using 3D convolution in the video as opposed to an LSTM and show similar results on the standard sensor fusion problem.

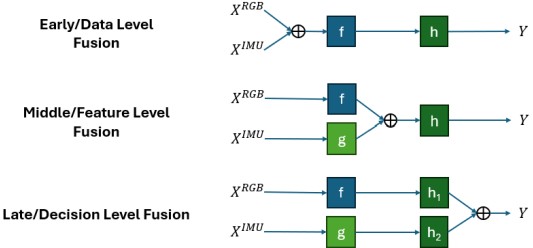

Figure 7: Types of Sensor Fusion

Sensor fusion is often broken down into the following 3 methods based on where the data are combined [33, 25, 34], also shown in Figure 7: 1) Early or data-level fusion combines the raw sensor outputs before

any processing. 2) Middle/intermediate or feature-level fusion combines each sensor modality after some preprocessing or feature extraction. 3) Late or decision-level fusion combines the raw output, essentially ensembling separate models.

The UTD-MHAD dataset has no predefined splits and benchmarks, and little to no works have open sourced their code on it. As such, to ensure we were using the data correctly implemented a few sensor fusion models and compared to state-of-the-art reported methods and showed similar performance results. The results are given in Table 8 These experiments provide a standard of comparison for our results with other methods on the UTD-MHAD dataset, and these models are available in the code for this paper.

Table 7: Data Splits

| Split Name | % of Data | Provided Data |
|---|---|---|
| Train a) | 40% | $(X^{RGB}, Y)$ |
| Train b) | 40% | $(X^{RGB}, X^{IMU})$ |
| Validation | 10% | $(X^{RGB}, X^{IMU}, Y)$ |
| Test | 10% | $(X^{RGB}, X^{IMU}, Y)$ |

Table 8: SOTA Sensor Fusion Performance on UTD-MHAD

| REPORTED MODELS | ACCURACY |
|---|---|
| HAMLET [17] | 95.12% |
| WEI ET AL. [42] | 95.6% |

| REPRODUCED FROM [42] | ACCURACY |
|---|---|
| EARLY FUSION | 86.71% |
| FEATURE FUSION | 95.60% |
| LATE FUSION | 94.22% |

### D.2.3  Contrastive Learning Baseline

ImageBind [13] learns encoders for 6 modalities, (Images/Videos, Text, Audio, Depth, Thermal and IMU) by performing CLIP's training method [32] between each of those encoders and the Image/Video encoder. It was well tested for image, text and audio based alignment, retrieval and latent space generation tasks, however was not well test with IMU data and not used for specific tasks, such as HAR. In addition, one fundamental difference between Imagebind and CA is that Imagebind constructs a latent space amongst the sensing modalities and text and aligns between them. We hypothesize that this is vector space is more difficult and unnecessary to construct, for human action recognitoin using sensing modalities. The text modality, although sequential in nature, does not have a time dimension, thus it cannot leverage correlations between modalities in time like C3T.

Let's denote the video, IMU and text encoders as $g^{(1)} : \mathcal{X}^{(1)} \to \mathcal{Z}, g^{(2)} : \mathcal{X}^{(2)} \to \mathcal{Z},$ and $g^{(3)} : \mathcal{X}^{(3)} \to \mathcal{Z}$ respectively. We perform two conventional task-specific adaptations for CLIP models. First, we attempt zero-shot transfer, in which we pass all the action labels through the text encoder. For a dataset with $C$ classes, we have $\hat{Z}^{(3)} = (\hat{z}_1^{(3)} \dots \hat{z}_C^{(3)})$. Finally, for a given IMU sample $(x_i^{(2)}, y_i) \in \mathcal{D}_{Test}$, we pass $x_i^{(2)}$ through the IMU encoder $g^{(2)}$ and retrieve $\hat{z}^{(2)}$. Finally, we classify the point by looking at which points gives the highest cosine similarity score in the latent space, e.g. $\hat{y}_i = argmax_j \frac{\langle x_i^{(2)}, \hat{z}_j^{(3)} \rangle}{||\langle x_i^{(2)}, \hat{z}_j^{(3)} \rangle||}$.

Given that ImageBind is a large model trained on massive corpuses of data it becomes impractical to train the model from scratch on our smaller datasets collected from wearables and edge devices. Instead, we fine-tuned the ImageBind model using a linear projection head on the encoders, that can then be trained for a specific task. The results of this method are depicted in Table 6.

The results show a poor generalization of Imagebind to most experiments on our setup, and we hypothesize a few reasons. Firstly, ImageBind is a large model and may either overfit to small datasets, or not have enough training examples to learn strong enough representations. Second, ImageBind was pre-trained on Ego4D and Aria which contain egocentric videos to align noisy captions with the IMU data, whereas our datasets had fixed labels and were mostly 3rd person perspective. In fact ImageBind performed the best on the one egocentric dataset we used, MMEA-CL[43]. Lastly, Imagebind was trained on a IMU sequences of 10s length sampled at a much higher frequency, thus we zero-padded or upsampled the IMU data to fit into ImageBind's IMU encoder, and the sparse or repetitive signal may have been too weak for ImageBind's encoder to accurately interpret the data.

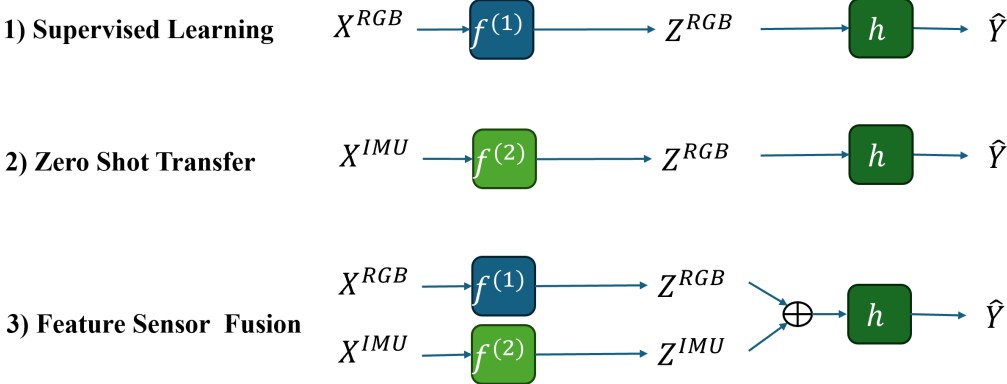

Figure 8: We can perform inference with the model with any subset of it's original input modalities.

# E   Additional Experimental Details

We have shown that the CA method performs well on a new modality, however, the question remains whether it can still retain performance on the original modality it was trained on. Furthermore, if it is given multiple modalities during inference, can it leverage information from all of them? Through CA, for any data sample $i$ given for inference, regardless of whether the sample contains data from $\mathcal{X}^{(1)}$, $\mathcal{X}^{(2)}$ or both $(\mathcal{X}^{(1)}, \mathcal{X}^{(2)})$, we can estimate the latent vector $\hat{z}_i \in \mathcal{Z}$ and thus predict the action using the HAR module, $\hat{y} = h(\hat{z}_i)$

**1. RGB (Supervised Learning)** The typical supervised machine learning paradigm tests the model on different samples of the same distribution. In our case, this is testing CA on RGB data. Thus the estimated latent vector is given by $f^{(1)}(x_i^{(1)}) = \hat{z}_i$.

**2. IMU (Zero-Shot Transfer)** The cross-modal zero-shot transfer method is the main result of this paper and described above in Appendix C. Here the estimated latent vector is given by $f^{(2)}(x_i^{(2)}) = \hat{z}_i$.

**3. Fusion (Sensor Fusion)** When both modalities are present, the model estimates the latent vector $\hat{z}_i$ from the outputs of modality-specific encoders assuming each estimate is equally as good as the other. $\hat{z}_i = \mathbb{E}[z_i | x_i^{(1)}, x_i^{(2)}] = E[z_i | \hat{z}_i^{(1)}, \hat{z}_i^{(2)}] = \frac{\hat{z}_i^{(1)} + \hat{z}_i^{(2)}}{2}$ . Thus, for sensor fusion we average the outputs of each of the encoders.

# F   Related Works

## F.1   Sensor Fusion

Cross-modal transfer learning is a method to transfer knowledge from a modality with abundant training data to one with limited data [28]. Cross modal transfer is a subset of domain adaptation. Domain adaptation allows a machine learning model trained in one domain to efficiently adapt to another related domain for the same output task with fewer data labels [29, 10]. Given this focus on scarcely labeled domains, adaptation is often performed through unsupervised [6] or semi-supervised [1] methods. In terms of human activity recognition, different data domains can imply adapting between different sensor inputs [2], different positions of wearables on the human body [38, 6, 31], different users [16, 12] or IMU device type [20, 45, 5]. In the context of this cross-modal learning our work may be considered domain adaptation involving different sensor inputs.

**IMU Virtualization:** To overcome the difficulties associated with limited IMU data, many recent works have leveraged the ability to simulate IMU data from videos such as IMUTube [22] or ChromoSim [15]. This allows them to train an IMU model from camera data and perform zero-shot classification on IMU data with a model that has never seen real IMU data. However, these models are time and data-intensive and cannot easily be extended to other modalities.

**Contrastive Learning Methods:** Contrastive learning-based multimodal models such as ImageBind [13] or IMU2CLIP [27] are relatively easy to extend to new modalities, however, they fail to fuse the sensor modalities when present and are designed for cross-modal retrieval or generation and perform poorly on specific tasks such as human action recognition.

**Cross-Modal Knowledge Distillation** Knowledge distillation methods typically use an extra auxiliary modality during training to increase single modal performance during testing, however, they assume labeled training data from both modalities during training [44, 21, 39, 3]. Notably, [36] attempts to perform without knowledge distillation without labels for one modality using a student-teacher framework which CA can out-perform. Furthermore, they test transferring acrosss visual modalities which is likely more correlated that visual and inertial modalities.

**Unsupervised Modality Adaptation:** Unsupervised Modality Adaptation is a new term we coined, however, previous methods have explored similar settings. Thoker and Gall [36] tests on a modality that doesn't seem labels during training, however, they work with RGB and skeletal modalities, both are which derived from visual modalities, making their task inherently easier. Their work is also limited to one dataset, tested with only an ensemble based student-teacher method, and their code and model is not opensourced.

Furthermore, MMG-EGO-4D [14] provides a very similar zero-shot cross-modal transfer setting and they also work with IMU and RGB data. However, their work is limited to one custom-built egocentric dataset, and they only analyze one method (similar to our CA method). Their method also focuses more on 'few-shot' output generalization to new classes as opposed to latent space alignment techniques for cross-modal UMA.

