# OpenReview forum: "Unsupervised Modality Adaptation in Human Action Recognition via Cross-modal Representation Learning"
_NeurIPS.cc/2024/Workshop/UniReps — UniReps_

### Official Review · Reviewer_cfvd · 2024-09-29
**Evaluation of the paper "Unsupervised Modal Adaptation in Human Behavior Recognition through Cross Modal Representation Learning"**

**Rating:** 8
**Confidence:** 4

**Review:**

This article introduces the concept of Unsupervised Modal Adaptation (UMA) in Human Behavior Recognition (HAR) by using a unified multimodal representation space to transfer knowledge between modalities. The author proposes three UMA methods: Student Teacher (ST), Contrastive Alignment (CA), and Cross Modal Time Transfer (C3T). These methods were tested on multimodal datasets, including IMU and RGB data. This article emphasizes the unique ability of C3T to handle temporal noise and misalignment, demonstrating its effectiveness in noisy real-world scenarios.

---

> ### Author Response · Authors · 2024-11-07
>
> We deeply appreciate your positive review of this work!

---

### Official Review · Reviewer_rLW3 · 2024-10-05
**C3T is novel for Unsupervised Modality Adaptation in action recognition**

**Rating:** 7
**Confidence:** 4

**Review:**

I. paper summary:

This paper conducted a thorough study on Unsupervised Modality Adaptation in action recognition by utilizing RGB+IMU multi-modality representation aligning.  Since generally the modality of video data are RGB information from camera, the other modalities like IMU lacks human annotations. The authors explored three different methods to transfer the action recognition capability from RGB modality to IMU modality, namely Student-Teacher (ST), Contrastive Alignment (CA), and Cross-modal Transfer Through Time (C3T). In particular, the novel C3T demonstrate unique robustness on dealing time-varying latent vectors for time series data.

II. pros:

1. The task of transferring knowledge from RGB modality to IMU modality seems interesting and practical in real world applications.  The proposed method on aligning multi-modalities is closely related to the topic of  UniReps workshop.

2. The proposed C3T contains novelty.  Built on the encoders trained by contrastive learning, C3T measures the frame-wise  similarity of RGB embeddings  and IMU embeddings.  Then a transformer structure is utilized to merge and learn from these two modalities.

3.  The experiments are concrete and persuasive,  demonstrating the advantages and disadvantages of ST, CA, C3T.




III. cons and questions:

1. the ST methods is a variant of knowledge distillation with pseudo labels in action recognition domain,  which has been explored in a work 'Knowledge integration networks for action recognition' in AAAI 2020. It would be great if the authors could compare the differences of the proposed method and 'Knowledge integration networks for action recognition'.

2. The experiments have demonstrated that the proposed C3T is not always better than CA and ST.  While it is good to explore the different properties of these three methods, is it possible to improve the disadvantages of C3T?

3. The proposed C3T is capable of dealing with time-varying latents and robust to time noise. So what is your sampling strategy? Do you utilize video-level sampling strategy like TSN (ECCV 2016) and TFCNet (arxiv 2022), or clip-level sampling like 3D CNNs, eg. Nonlocal networks ( CVPR 2018), R2+1D (CVPR 2018) , Slowfast (ICCV 2019)? Or hybrid sampling like V4D (ICLR 2020)? Generally video-level sampling strategy and hybrid sampling will be more robust. Maybe the authors have tried these methods. I suggest the authors to compare these sampling methods in the experiment.

---

> ### Author Response · Authors · 2024-11-07
>
> We sincerely thank the reviewer for their in-depth analysis of our work and their positive outlook.  We would like to respond to the listed Cons/Questions and invite the reviewer to follow up further with any thoughts /questions.
>
> III. Cons and Questions:
> 1. It seems the cited work indeed uses a similar student-teacher network, however, Knowledge Integration in that work refers to knowledge integration across auxiliary tasks as opposed to various modalities and that work is limited to video human action recognition (single modality). We provide a slightly more extensive related work in Appendix F where we discuss other cross-modal knowledge distillation methods and in particular methods that work with RGB and IMU data.
> 2. Yes. We indeed have updated our experiments (now we perform 3 trials and take the average) and we observed that on average C3T does indeed perform better. We also highlight circumstances where C3T always performs better (with time shift noise).
> 3. This is a great point! We are using a simple video-level sampling, and have not yet conducted a comprehensive comparison of different sampling strategies for C3T. Your suggestion to compare video-level sampling (like TSN and TFCNet), clip-level sampling (like NonLocal Networks, R2+1D, SlowFast), and hybrid sampling (like V4D) is excellent and would provide valuable insights. Although we did not have the capacity to perform these experiments at the moment, we will investigate this further for future iterations of this work.

---

### Official Review · Reviewer_kDXS · 2024-10-06

**Rating:** 6
**Confidence:** 3

**Review:**

## Summary
The paper introduces a novel approach to human action recognition (HAR) called Unsupervised Modality Adaptation (UMA), which aims to transfer knowledge between modalities without labeled data for the target modality during training. The authors propose three methods for UMA: Student-Teacher (ST), Contrastive Alignment (CA), and Cross-modal Transfer Through Time (C3T). Each method leverages a unified multimodal representation space to enable cross-modal knowledge transfer. The effectiveness of these methods is demonstrated through extensive experiments on various datasets combining RGB video and Inertial Measurement Unit (IMU) data.

## Strengths
1. Three different methods (ST, CA and C3T) are proposed, and the advantages and disadvantages of these methods are compared and analyzed in detail.
2. The methods are tested on four different datasets, demonstrating their effectiveness across various conditions and data types.

## Weaknesses
1. This paper mainly focuses on RGB and IMU data. It would be better to discuss other types of multimodal data, such as image-text.
2. There is a lack of visualization and a lack of discussion of the successful and failure predictions.
3. The effect of hyperparameters on performance should be discussed.

---

> ### Author Response · Authors · 2024-11-07
>
> Thank you for the insightful feedback. We would like to respond to the listed weaknesses and invite the reviewer to follow up further with any thoughts /questions.
>
> Weaknesses:
> 1. We acknowledge this work is limited to RGB and IMU data. We believe that the C3T method might not bring any unique advantage in cross-modal alignment with Text as a modality, as text is discrete, not a time-continuous signal. Although, with some modification, this may be an interesting direction to pursue.
> 2. We have performed some visualizations using TSNE on the latent representations (Figure 3) and the C3T Attention head weights (Figure 4). These were added to the appendix, Figure 3.
> 3. We Have added some ablations in Tables 4 and 5  of the appendix, as well as an analysis of the latent vector "hidden_size" hyperparameter. This analysis shows convolutions are indeed the best architecture for the encoders and C3T maintains a strong performance even with smaller model sizes.

---

### Official Review · Reviewer_Gguh · 2024-10-06
**Unsupervised Modality adaptation in HAR via Cross-Modal Representation**

**Rating:** 7
**Confidence:** 5

**Review:**

The paper proposes an effective mechanism to align multiple modalities or sensors proposing three techniques. However unsupervised alignment does exist (perhaps not in HAR) so it would be good if the authors could perhaps frame the paper if this is specific to HAR or could be aligned to other modalities.



[1]Generalized Unsupervised Manifold Alignment

---

> ### Author Response · Authors · 2024-11-07
>
> We sincerely appreciate your positive review. Your observation that connects our work to generalized unsupervised manifold alignment is particularly valuable. While space constraints and the specific focus of this paper precluded an in-depth discussion or reframing of our work in this context, we certainly intend to explore this connection more thoroughly in future iterations of our research!

---

### Decision · Program_Chairs · 2024-10-10

**Decision:**

Accept

**Comment:**

In light of the positive reviewers' feedback and relevancy of the submission, we are pleased to accept this paper for presentation at UniReps 2024. We kindly ask the authors to incorporate the reviewers' suggestions and feedback in the final camera-ready version of the manuscript.

---

> ### Author Response · Authors · 2024-11-07
>
> We would like to thank all the reviewers for their feedback and positive reviews. We have updated the paper with a few changes:
>
> 1. We replaced Table 2 with more robust experiments (each experiment was repeated 3 times and the average is displayed). The overall trends are relatively the same as before. We also compare performance in Unsupervised Modality Adaptation to the performance of the same modules in a Supervised setting. The original table with the sensor fusion and imagebind baseline are in the appendix for reference (Table 6), but believe that this new table 2 is more informative and the supervised baseline is more competitive/interesting.
>
> 2. We condensed the other tables into one Table 1, and similarly repeated the experiments thrice and averaged their results, for more robust empirical results.
>
> 3. We added some strong visualizations to support our method (Figures 3 and 4), some ablations (Table 4 and 5) and a computational efficiency analysis (Figure 5).